# Comparison of Twelve Ant Species and Their Susceptibility to Fungal Infection

**DOI:** 10.3390/insects10090271

**Published:** 2019-08-26

**Authors:** Nick Bos, Viljami Kankaanpää-Kukkonen, Dalial Freitak, Dimitri Stucki, Liselotte Sundström

**Affiliations:** 1Organismal and evolutionary biology, University of Helsinki, 00100 Helsinki, Finland; 2Tvärminne Zoological Station, 10900 Hanko, Finland; 3Section for Ecology & Evolution, University of Copenhagen, 2200 Copenhagen, Denmark; 4Faculty of Social Sciences, University of Helsinki, 00100 Helsinki, Finland; 5Institute of Biology, University of Graz, 8010 Graz, Austria

**Keywords:** behavior, grooming, exposure, infection, metapleural gland, mortality, pathogen

## Abstract

Eusocial insects, such as ants, have access to complex disease defenses both at the individual, and at the colony level. However, different species may be exposed to different diseases, and/or deploy different methods of coping with disease. Here, we studied and compared survival after fungal exposure in 12 species of ants, all of which inhabit similar habitats. We exposed the ants to two entomopathogenic fungi (*Beauveria bassiana* and *Metarhizium brunneum*), and measured how exposure to these fungi influenced survival. We furthermore recorded hygienic behaviors, such as autogrooming, allogrooming and trophallaxis, during the days after exposure. We found strong differences in autogrooming behavior between the species, but none of the study species performed extensive allogrooming or trophallaxis under the experimental conditions. Furthermore, we discuss the possible importance of the metapleural gland, and how the secondary loss of this gland in the genus *Camponotus* could favor a stronger behavioral response against pathogen threats.

## 1. Introduction

Ants are ubiquitous in most terrestrial environments [1]. However, the underground environment in which they live is also home to a diverse community of microbes [2]; thus, ants get exposed to a plethora of potential pathogens present in and on the soil. These pathogens can pose a problem when brought inside the nest, as the dense population inside a nest increases the chance of the rapid spread of parasites and disease, a risk that may be aggravated by the high relatedness among individuals within colonies [3,4,5,6].

To cope with the challenges an increased pathogen pressure poses, ants employ a variety of strategies to prevent infections from becoming established. These range from individual behavioral and physiological responses, to collective behaviors that convey disease control, referred to as “social immunity” [3,7,8,9,10]. Individual behavioral responses may include avoidance of the pathogen, which can be achieved either by avoiding areas that contain pathogens [11,12] or by avoiding the consumption of contaminated food [9]. Another example is autogrooming, whereby an individual cleans its own cuticle in order to remove pathogens [13]. Many species of ants also have a metapleural gland, which can produce antimicrobial secretions to aid in disease defense [14,15]. The metapleural gland can be considered part of an individual’s immune system, whereby an individual can spread the acidic secretions over its cuticle, and so contain an infection [16]. 

Social insects can also contain diseases via social interactions. Thus, in some cases, the metapleural gland can be thought of as part of the social immune system. For example, leaf cutter ants are known to spread the secretions onto their fungiculture, thereby protecting the colony as a whole [14,16]. Many other social immune defenses have been described. For example, wood ants collect antimicrobial resin from their environment in order to help fight pathogens [17,18,19]. Workers can groom nestmates, and thereby more efficiently remove pathogens (allogrooming, [20,21]), individuals can exchange food through mouth-to-mouth feeding (trophallaxis), which has been shown to allow the transfer of antimicrobial substances that increase a receiver’s resistance to infection [22]. Infected individuals have been shown to leave the nest on their own accord to die in isolation [23,24], and infected brood can be removed through destructive disinfection [25]. Dead individuals can be carried out of the nest (necrophoresis, [26,27,28]), and even be buried [29]. Cannibalism of infected workers has been shown to occur in termites [30]. Lastly, hygienic behavior in the form of waste management has been described in fungus-growing ants [31,32]. The responses and adaptations listed above should be effective against a broad range of pathogens, regardless of the species of the pathogen [16,33]. 

To better understand variability between ant species in their susceptibility to pathogens, as well as how different pathogens affect the ant species, we investigated how two opportunistic pathogenic fungi (*Beauveria bassiana* and *Metarhizium brunneum*) affect the survival and behavior of twelve ant species from four different genera. The ant species were all common in the study area, which was characterized by the oligotrophic, moderately acidic soils that are typical of coniferous forests [34], as well as habitats consisting of pine and spruce stands, dry meadows, and lusher, grove-like patches [35]. Following exposure to the pathogens, we screened for three specific behaviors (autogrooming, allogrooming, trophallaxis), which previously have been associated with pathogen exposure (e.g., [20,21,22,24,36,37]). Although the use of generalist pathogens has been criticized in studies of social immunity [38], the use of specialized pathogens (e.g., *Pandora formicae*, [39]) would be counterproductive in this context, as specialized pathogens would prevent a multi-species comparison due to their narrow host range. 

## 2. Materials and Methods 

### 2.1. Collection and Housing

Twelve species of ants (*Lasius platythorax*, *Lasius flavus*, *Camponotus ligniperdus*, *Formica truncorum*, *Formica exsecta*, *Formica lugubris*, *Formica sanguinea*, *Formica pratensis*, *Formica fusca*, *Formica cinerea*, *Myrmica schencki* and *Myrmica ruginodis*) were collected in the Hanko archipelago area, SW Finland, in the spring of 2013. The species are common on the islands around Tvärminne Zoological Station, co-occur in the same habitat patches, are ground-dwelling, and have roughly the same diet (based on honeydew and insect prey) [40,41]. Most species nest in soil; however, four of the Formica species (*F. truncorum*, *F. pratensis*, *F. lugubris*, and *F. exsecta*) build a mound above their underground chambers, which usually comprises spruce needles and twigs. *Lasius flavus* is hypogeic, and uses mainly root aphids for nutrition [42]. For each of the 12 species, workers of three colonies were collected. No attempt was made to collect colony queens, given that this would lead to the destruction of the colony, and because we would not have been able to obtain queens for all colony fragment for the monogyne species. The ants were maintained in the laboratory only for the duration of the experiment (15 days), hence we assumed the absence of a queen would have little or no impact on worker behavior. The workers were all fully sclerotized, and had hibernated since the previous summer. Each collected colony was housed in a large nest box (the walls of which were coated with Fluon (Whitford) to prevent the ants from escaping), containing the colony fragment’s own nest material. Here the ants were allowed to habituate for one day. The next day, 180 workers from each colony were collected at random, and divided across six jars (7 cm Ø × 5 cm), the walls of which were coated with Fluon, with bottoms lined with plaster to maintain moisture and facilitate movement for the ants. In the case of *Camponotus ligniperdus*, which has a strong morphological caste differentiation, only small-to-medium workers (non-soldiers) were used. Each jar was closed with a plastic lid containing six small openings for ventilation. In total, we had six jars per colony, each containing 30 workers (180 workers total). With three colonies per species, this resulted in 18 jars per species, 216 jars in total (6480 ants), divided over 12 species. Throughout the experiment, ants were fed ad libitum according to a Bhatkar–Whitcomb diet [43]. Furthermore, an Eppendorf tube filled with water and a cotton plug was provided as a source of water, as well as to maintain ~70% humidity. Jars were kept at room temperature (~20 °C) throughout the experiment.

### 2.2. Procedures: Species-Specific Survival

To assess the survival of ants upon exposure to pathogenic fungi, conidia of two fungal pathogens (*Beauveria bassiana* and *Metarhizium brunneum*) were harvested from SDA plates using 10 mL 0.01% Triton-X, following the methodology of Bos et al. [24], and diluted to a final concentration of 10^8^ conidia/mL. Viability of conidia was checked by plating the conidia on SDA agar plates, and incubating them at 23 °C overnight. The plates were checked the next day for hyphal growth. Viability of conidia was >95% in all cases. The strains used for this study were KVL 04–57 for *M. brunneum,* and KVL 03–90 for *B. bassiana* (provided by the Department of Plant and Environmental Sciences, University of Copenhagen). Danish strains were used to prevent potential local adaptation from influencing the results.

The six jars per colony were divided across three treatments (two jars per treatment per colony): (1) Triton-X control; (2) exposure to *B. bassiana*, and (3) exposure to *M. brunneum.* To expose the ants to either the control (0.01% Triton-X), or one of the two fungal treatments, and to ensure full coverage of the body, individual ants were submerged into the solution for 5 s, after which excess liquid was removed by placing the ant on clean absorbent tissue paper. Afterwards, each individual was returned to its assigned jar. Mortality was assessed daily for 14 days. On day four, 10 ants per colony were removed for a separate experiment, and thus added as censored data in the datafile. In four cases (*M. ruginodis–B. bassiana* colony 3, replicate 2; *F. truncorum*–control colony 3, replicate 1; *L. platythorax–B. bassiana* colony 1, replicate 1; and *L. platythorax*–control colony 3, replicate 2) the jars were not coated sufficiently with fluon, and/or the lids were defective, resulting in many ants escaping over time. These pots are therefore excluded from the survival analysis. 

### 2.3. Procedures: Behavioral Measurements

During the first three days after treatment, two 3-s videos from each jar were recorded 10 min apart, twice per day (morning and afternoon), for a total of four videos per day. We took 3 s videos instead of pictures, to more reliably assess expressed behaviors. All 2588 videos were watched by one observer, who was blind with respect to treatment. From each video, the total number of ants performing autogrooming (including acidopore grooming), allogrooming, and trophallaxis were recorded.

### 2.4. Procedures: Effect of the Metapleural Gland on Survival

During the survival experiment, we found a strong difference in mortality between infected *C. ligniperdus* workers (who do not have a metapleural gland) and the other ant species. We therefore also assessed the potential role of the metapleural gland in survival following fungal infection. For this we selected one species of the genus *Formica* (*F. sanguinea*) that exhibited moderate survival upon exposure to *B. bassiana* in the main experiment, allowing us to detect effects in both directions (higher vs lower survival). We thus divided 80 individuals from each of the three colonies of *F. sanguinea* into four jars per colony (20 individuals/jar). The jars were assigned to one of two treatments. In one of these treatments (henceforth called Blocked), the metapleural glands of each individual ant were blocked using enamel paint, whereas the control jars (henceforth called Control) were handled identically, but without enamel paint. The ants were then left alone for five days, to reduce the probability of fresh metapleural gland secretions being present on the cuticle of ants with a blocked metapleural gland. After the five days, the ants were subdivided into two further treatments: exposure to Triton-X (Trix), and exposure to 10^8^ conidia/mL of *B. bassiana* (BB). The experimental design thus encompassed four treatments (Trix-Control, BB-Control, Trix-Blocked, and BB-Blocked). 

### 2.5. Statistical Analysis

All analyses were performed in R Version 3.5.1 [44,45], using the packages survival [46], multcomp [47], MASS [48], broom [49], car [50], vcd [51], and plyr [52]. For all statistical tests we used a significance threshold of α = 0.05.

#### 2.5.1. Survival

Survival of each species was analyzed using a parametric survival regression with Weibull distribution via the survreg function from the survival package. Treatment was added as a fixed effect, with Replicate nested within Colony, to control for pseudoreplication. As we had fewer than five levels within both Colony and Replicate, we modeled these as fixed (instead of random) effects. This rendered the model more conservative. As the model only compares the control of the two fungal treatments, planned contrasts between the fungal treatments themselves were obtained by releveling the factor using the relevel function. Individuals that did not die during the experiment were entered as censored data.

#### 2.5.2. Behavior

Self-grooming followed a negative binomial distribution in all species except *F. exsecta,* in which a Poisson distribution gave a better fit (distribution was checked using the goodfit function of the vcd package). For all species we used a generalized linear model with Treatment as a fixed effect. Day, as well as Replicate nested within Colony, were also added as fixed effects in order to account for pseudoreplication. Adding these as random effects was not possible because we had fewer than five levels. All species, except *F. exsecta*, were analyzed using the glm.nb function; *F. exsecta* was analyzed using the glm function specified with a Poisson family. As with the survival analysis, the two treatments (*B. bassiana* and *M. brunneum*) were compared using the relevel function. Most species showed little or no allogrooming (mean = 0.06 events per recording), and trophallaxis (mean = 0.06 events per recording) in all treatments, which prevented a proper analysis. We also tested whether survival upon pathogen exposure was correlated with the amount of autogrooming across species. For this, we defined survival as the proportion of ants still alive after day 14. The amount of autogrooming per replicate was defined as the mean number of autogrooming events across all sampling points. Then, we constructed a linear model with the log+1 of autogrooming as the dependent variable (to improve homoscedasticity). Survival and treatment (*B. bassiana* vs *M. brunneum*), as well as the interaction between these, were added as fixed effects, and colony nested within species was added as a fixed effect, to account for pseudoreplication. 

#### 2.5.3. Metapleural Gland

The survival of *F. sanguinea* workers in the metapleural gland experiment was analyzed using a parametric survival regression with Weibull distribution, using the survreg function. The model included fungal treatment (Trix vs BB), metapleural gland treatment (Control vs Blocked), and Colony as fixed effects, as well as the interaction between them. Four comparisons were pre-planned (Trix–Control vs Trix–Blocked; BB–Control vs BB–Blocked; Trix–Control vs BB–Control and Trix–Blocked vs BB–Blocked). These were analyzed using the glht function of the multcomp package. We controlled for multiple comparisons by setting the test parameter to adjusted (“fdr”) [53].

## 3. Results

Exposure to *B. bassiana* or *M. brunneum* both led to a significant decrease in survival in all species, except *L. platythorax*, the survival of which decreased significantly only when exposed to *B. bassiana* (Table 1, Figure 1). Furthermore, infection by *B. bassiana* led to a significantly higher mortality than *M. brunneum* in all species (Table 1, Figure 1). However, the degree to which species were susceptible to infection differed extensively, with *C. ligniperdus* extremely susceptible to both fungi, whereas *M. schencki*, for example, showed minimal mortality (Figure 1). 

Overall, trophallaxis occurred at a rate of 0.06 ± 0.24 (mean ± SD) events per recording. This amount ranged from 0.00 ± 0.00 in *M. schencki* to 0.19 ± 0.45 in *F. cinerea*. Allogrooming occurred at a rate of 0.06 ± 0.31 (mean ± SD) events per recording, ranging from 0.00 ± 0.07 in *F. fusca* to 0.40 ± 0.78 in *F. cinerea*. Due to the low occurrence, these behaviors were not statistically analyzed. The summarized data per treatment and species can be found in Appendix A (allogrooming) and Appendix A (trophallaxis).

Autogrooming occurred at rates between 0 and 7 per time unit among unexposed ants, and increased significantly upon exposure to fungus in all species except for *L. flavus* and *L. platythorax* (Table 2, Figure 2). However, in *M. schencki*, this increase was only significant when exposed to *M. brunneum*, and in *F. lugubris* only when exposed to *B. bassiana*. Compared to the controls, autogrooming significantly decreased in *F. lugubris* and *L. flavus* when exposed *M. brunneum,* and in *L. platythorax* when exposed to *B. bassiana* (Table 2, Figure 2). Species with higher survival generally groomed less (lm, survival, F = 5.81, *p* = 0.02, Figure 3), regardless of the pathogen used (lm, survival × treatment, F = 2.00, *p* = 0.28).

Blocking the metapleural gland of *F. sanguinea* workers did not result in an increased mortality in the unexposed control group (Trix–Blocked vs Trix–Control, z = 0.021, *p* = 0.98), but exposure to *B. bassiana* significantly increased mortality, regardless of whether the metapleural gland was blocked or not (Trix–Blocked vs BB–Blocked, z = 4.63, *p* < 0.01; Trix–Control vs BB–Control, z = 3.36, *p* < 0.01). However, the rate of survival did not differ significantly across exposed ants with (versus without) their metapleural glands blocked (BB–Blocked vs BB–Control, z = −1.98, *p* = 0.06, Figure 4). 

## 4. Discussion

We studied the effects of exposure to two different pathogens (*M. brunneum* and *B. bassiana*) on the survival and behavior of twelve species of ants belonging to four genera, and found pronounced differences among the investigated ant species in their susceptibility to the two pathogens. In addition, the virulence of the two fungal species differed across the ant species. The level of allogrooming and trophallaxis was very low in all species, whereas autogrooming occurred at higher rates and increased in most species following exposure to pathogens. 

*C. ligniperdus* was the most susceptible to the pathogens, whereas the two *Myrmica* species were the least susceptible. The high susceptibility of *C. ligniperdus* could be due to the lack of a metapleural gland; however, the survival of *F. sanguinea* individuals with blocked glands was not significantly lower than that of individuals without blocked glands. Furthermore, *F. sanguinea* individuals with blocked metapleural glands had higher survival than *C. ligniperdus* when exposed to pathogens. Thus, it is clear that the lack of a metapleural gland was not the only factor contributing to the lower survival of *C. ligniperdus.*


In the genera with multiple representatives (*Lasius*, *Myrmica*, and *Formica*), susceptibility varied extensively. Variation in susceptibility to pathogenic fungi may be partly attributable to the phylogenetic background. Palearctic *Formica* include several subgenera [54], of which *F. truncorum, F. pratensis,* and *F. lugubris* are placed in the subgenus *Formica s. str*. These three species show very similar survival curves. However, of the remaining *Formica* species, all but two belong to different subgenera—*Raptiformica* (*F. sanguinea*), *Coptoformica* (*F. exsecta*), and *Serviformica* (*F. fusca* and *F. cinerea*)—preventing detailed phylogenetic comparisons. All study species were collected from similar habitats in the same area, are ground dwelling, and overlap in diet (scavenging and aphid tending). However, four of the *Formica* species (*F. truncorum, F. pratensis, F. lugubris*, and *F. exsecta*) are mound-building, and all appear to have very similar survival curves. Interestingly, these four species were among the least resistant towards both *M. brunneum* and *B. bassiana*. The question remains whether these similarities were due to the close phylogenetic relationship among these species, or due to similarities in their nesting habits (i.e., the microbial communities of their nest mounds). 

The differences in survival across ant species could be mediated by different methods of fighting infection. We recorded 30 ants per jar, four times per day, over the first three days after exposure. This method of recording ensured detection of any reasonably frequent behaviors. Nonetheless, trophallaxis, by which ants can transfer antimicrobial substances [22], appeared to occur very rarely. Trophallaxis has been associated with social immunity, and is normally measured when a treated ant is presented to non-treated nestmates [22,36,37]. Hamilton et al. [22] found an increase in trophallaxis after exposure to pathogens in *Camponotus pennsylvanicus*, but Konrad et al. [37] did not find an effect of exposure in *Lasius neglectus*. In contrast, Aubert et al. [36] found a decrease in trophallaxis after *Formica polyctena* workers were immune-challenged with lipopolysaccharides (LPS). Unlike previous experiments [22,36,37], all ants in a jar were treated, and had access to food in our setup. This may explain the low occurrence of this behavior. 

Allogrooming was also observed only very rarely in any of the species. This is in agreement with results in *Formica selysi* [13]. Ants normally display allogrooming when detecting a nestmate, the cuticle of which has been manipulated or interfered with [37]. According to Konrad et al. [37], allogrooming increases both upon exposure to conidia, and exposure to a sham control [21]. As in the case of trophallaxis, the low rate of allogrooming probably reflects the fact that all workers in a jar were treated equally, instead of presenting a treated ant to untreated nestmates.

All species exhibited autogrooming. *C. ligniperdus* appeared to have the highest base level of this behavior, which is similar to the results found in a comparative study by Walker and Hughes [55]. However, the high level of autogrooming did not prevent mortality in our study, likely due to the high dose of fungus used (10^8^ conidia/m*L*). With a lower dose, the ants could potentially have prevented infection by removing enough conidia through grooming [13,56]. Yet, the high dose used here allowed us to test whether species that are more susceptible to pathogens on a physiological level (after conidia have penetrated the cuticle) generally show more autogrooming, without the removal of conidia affecting mortality, and thus biasing the results. Indeed, species that showed higher mortality upon infection also showed higher rates of grooming than those that appeared to be more resistant against the pathogens. This suggests that some species may rely more on behavioral adaptations, whereas others could rely more heavily on other defenses, such as metapleural gland secretions, physiological barriers, or even social immune responses (which we perhaps did not detect, due to our experimental setup). 

Interestingly, in one *Formica* species, and both *Lasius* species, exposure to pathogens appeared to reduce autogrooming. In *F. lugubris* and *L. flavus,* this effect was significant with *M. brunneum,* and in *L. platythorax* this was significant with *B. bassiana.* This agrees with results in *Lasius japonicus,* in which autogrooming decreased when exposed to high concentrations (10^7^ conidia/mL) of *Metarhizium anisopliae* [57]. However, this stands in contrast with a study on *L. neglectus* exposed to *M. anisopliae* [20], which found that autogrooming increased upon exposure to a high concentration of pathogen (10^9^ conidia/mL). The potential reasons for this decrease in autogrooming remain to be studied in more detail.

Not only did the ant species differ from one another in their susceptibility to the two fungal species, the fungi themselves differed in their virulence. Exposure to *B. bassiana* generally induced higher mortality than exposure to *M. brunneum*. This stands in contrast to results found in the termite *Coptotermes formosanus*, which suffered higher mortality from *M. anisopliae* than *B. bassiana* [33]. These differences in mortality may be attributable to general differences between the two fungi, or to the specific strains used in this project. In our experiment, the two fungi were compared at equal concentrations, which is not necessarily the case in nature. Indeed, *M. anisopliae* appears to be better able to persist in soil than *B. bassiana* [58]. If this also applies to *M. brunneum*, ants may be more likely to encounter higher concentrations of *M. brunneum* than *B. bassiana.* However, it is not only virulence that determines the success of a fungus, and other characteristics, such as sporulation, attachment to cuticle, and temperature tolerance, should also be considered [58,59]. 

## 5. Conclusions

In summary, we have shown that different species of ant differ significantly in their susceptibility to pathogens, even within the same genus. Species appear to differ in their tendency to autogroom, and those that groom more appear to be less able to fight the fungus after infection has established. This suggests that species may differ in their methods of fighting infection. For example, different species may engage different aspects of their immune system, such as efficient metapleural gland secretions [14,15], a tough cuticle [60,61], physiological responses [62], and/or behavioral responses [13,55]. Furthermore, the pathogens we used differed strongly in their virulence, such that *B. bassiana* generally induced higher mortality than *M. brunneum.* Yet, our comparative approach suggests that neither *B. bassiana* nor *M. brunneum* are major sources of mortality in ants in general [38,63], given that concentrations approximately six orders of magnitude higher than natural levels [64] did not result in 100% mortality. Indeed, some species (especially *Myrmica* sp. and *Lasius* sp.) only suffered minor effects of exposure, especially when exposed to *M. brunneum.* This is in agreement with the fact that there are no recorded cases of colonies being eradicated by the generalist pathogens used here [38]. Given the similarity in nesting environments of our study species, we did not consider ecological context in this experiment (as seen in [55,65]). Instead, we focused on species differences with similar ecology, with the possible exception of the mound-building *Formica* sp. A next step would be to measure differences under more natural protocols, such as by having ants forage through either sterilized or non-sterilized soil. Furthermore, conducting assays in a more natural setting would also take into account aspects of immune systems missed in this study, such as the benefits that some species could have from integrating antimicrobial resin into their nest [15,16,17,47,66].

## Figures and Tables

**Figure 1 insects-10-00271-f001:**
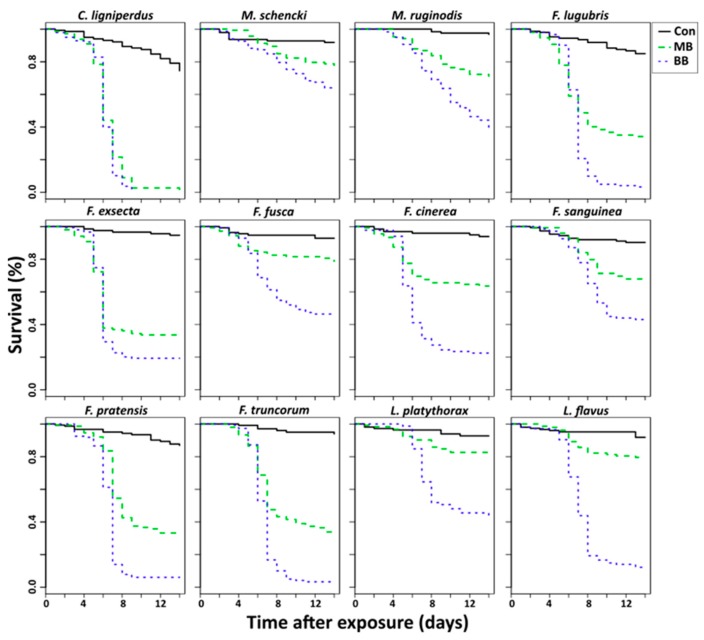
Survival comparisons between control ants (Con), ants exposed to *B. bassiana* (BB), and ants exposed to *M. brunneum* (MB).

**Figure 2 insects-10-00271-f002:**
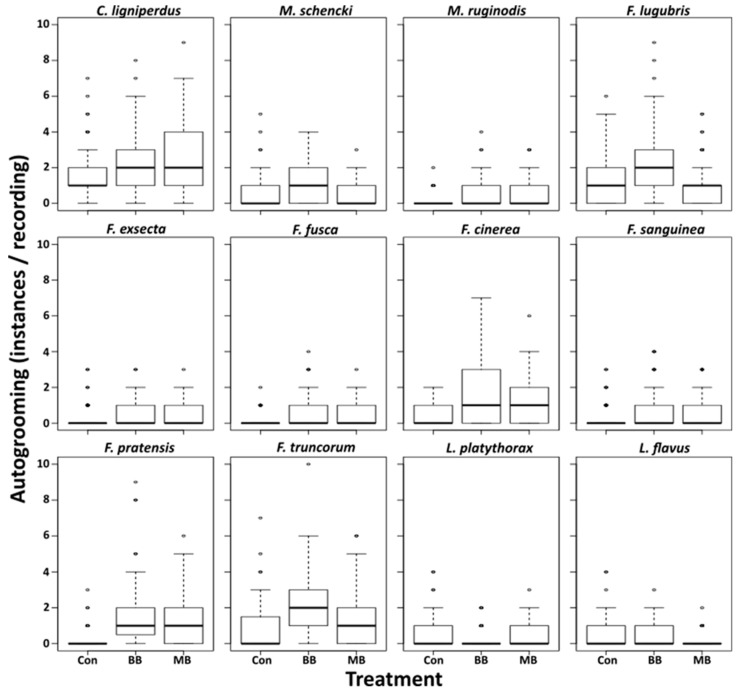
Autogrooming rates. Behavioral comparisons between control ants (Con), ants exposed to *B*. *bassiana* (BB), and ants exposed to *M. brunneum* (MB).

**Figure 3 insects-10-00271-f003:**
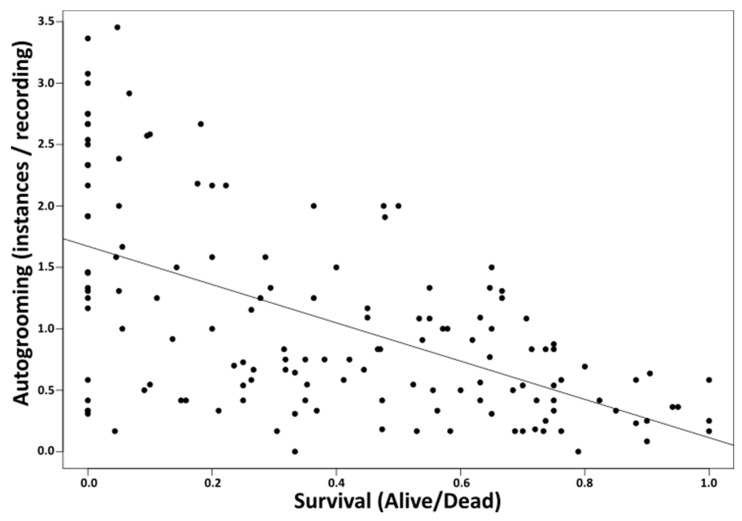
Autogrooming and survival. Colonies that show higher mortality upon exposure to pathogens express higher levels of autogrooming than colonies with higher survival.

**Figure 4 insects-10-00271-f004:**
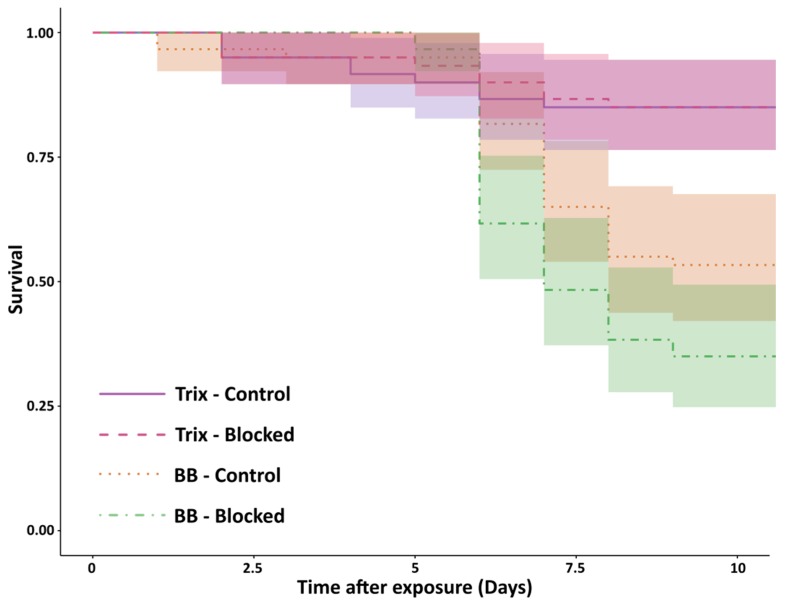
Effect of blocked metapleural glands on survival. Comparisons in survival between four treatments of *F*. *sanguinea* ants. Colored bands denote 95% confidence intervals.

**Table 1 insects-10-00271-t001:** Survival analysis. Comparisons in survival between control ants (Con), ants exposed to *B*. *bassiana* (BB), and ants exposed to *M. brunneum* (MB).

Species	Term	Estimate	Std. Error	Z	*p*-value
*C. ligniperdus*	Con-MB	−0.993	0.064	−15.513	<0.001
*C. ligniperdus*	Con-BB	−1.092	0.062	−17.493	<0.001
*C. ligniperdus*	MB-BB	−0.099	0.040	−2.489	0.013
*M. schencki*	Con-MB	−0.609	0.254	−2.399	0.016
*M. schencki*	Con-BB	−1.035	0.249	−4.158	<0.001
*M. schencki*	MB-BB	−0.426	0.175	−2.441	0.015
*M. ruginodis*	Con-MB	−1.098	0.261	−4.203	<0.001
*M. ruginodis*	Con-BB	−1.571	0.269	−5.834	<0.001
*M. ruginodis*	MB-BB	−0.473	0.110	−4.288	<0.001
*F. lugubris*	Con-MB	−0.876	0.110	−7.981	<0.001
*F. lugubris*	Con-BB	−1.317	0.107	−12.317	<0.001
*F. lugubris*	MB-BB	−0.441	0.059	−7.465	<0.001
*F. exsecta*	Con-MB	−1.556	0.215	−7.239	<0.001
*F. exsecta*	Con-BB	−1.905	0.213	−8.928	<0.001
*F. exsecta*	MB-BB	−0.348	0.073	−4.778	<0.001
*F. fusca*	Con-MB	−0.896	0.290	−3.092	0.002
*F. fusca*	Con-BB	−1.642	0.288	−5.695	<0.001
*F. fusca*	MB-BB	−0.747	0.177	−4.223	<0.001
*F. cinerea*	Con-MB	−1.316	0.274	−4.798	<0.001
*F. cinerea*	Con-BB	−2.097	0.277	−7.575	<0.001
*F. cinerea*	MB-BB	−0.781	0.127	−6.165	<0.001
*F. sanguinea*	Con-MB	−0.656	0.169	−3.875	<0.001
*F. sanguinea*	Con-BB	−1.075	0.170	−6.329	<0.001
*F. sanguinea*	MB-BB	−0.419	0.107	−3.930	<0.001
*F. pratensis*	Con-MB	−0.877	0.113	−7.785	<0.001
*F. pratensis*	Con-BB	−1.384	0.112	−12.410	<0.001
*F. pratensis*	MB-BB	−0.507	0.061	−8.373	<0.001
*F. truncorum*	Con-MB	−1.116	0.157	−7.125	<0.001
*F. truncorum*	Con-BB	−1.596	0.156	−10.260	<0.001
*F. truncorum*	MB-BB	−0.479	0.051	−9.382	<0.001
*L. platythorax*	Con-MB	−0.505	0.274	−1.844	0.065
*L. platythorax*	Con-BB	−1.290	0.275	−4.698	<0.001
*L. platythorax*	MB-BB	−0.784	0.184	−4.273	<0.001
*L. flavus*	Con-MB	−0.411	0.162	−2.540	0.011
*L. flavus*	Con-BB	−1.515	0.156	−9.679	<0.001
*L. flavus*	MB-BB	−1.104	0.109	−10.092	<0.001

**Table 2 insects-10-00271-t002:** Autogrooming. Comparisons of autogrooming behaviors between control ants (Con), ants exposed to *B. bassiana* (BB), and ants exposed to *M. brunneum* (MB).

Species	Term	Estimate	Std. Error	Z	*p*
*C. ligniperdus*	Con-MB	0.329	0.145	2.261	0.024
*C. ligniperdus*	Con-BB	0.351	0.145	2.417	0.016
*C. ligniperdus*	MB-BB	−0.023	0.136	−0.166	0.868
*M. schencki*	Con-MB	−0.045	0.229	−0.196	0.844
*M. schencki*	Con-BB	0.506	0.207	2.451	0.014
*M. schencki*	MB-BB	0.551	0.209	2.643	0.008
*M. ruginodis*	Con-MB	1.243	0.358	3.470	<0.001
*M. ruginodis*	Con-BB	1.277	0.357	3.581	<0.001
*M. ruginodis*	MB-BB	0.034	0.250	0.135	0.892
*F. lugubris*	Con-MB	−0.479	0.178	−2.690	0.007
*F. lugubris*	Con-BB	0.308	0.157	1.958	0.05
*F. lugubris*	MB-BB	0.787	0.173	4.543	<0.001
*F. exsecta*	Con-MB	0.712	0.261	2.725	0.006
*F. exsecta*	Con-BB	0.541	0.269	2.010	0.044
*F. exsecta*	MB-BB	−0.170	0.223	−0.764	0.445
*F. fusca*	Con-MB	1.646	0.400	4.113	<0.001
*F. fusca*	Con-BB	1.439	0.407	3.535	<0.001
*F. fusca*	MB-BB	−0.207	0.253	−0.818	0.414
*F. cinerea*	Con-MB	1.071	0.209	5.126	<0.001
*F. cinerea*	Con-BB	1.365	0.203	6.715	<0.001
*F. cinerea*	MB-BB	0.294	0.146	2.020	0.043
*F. sanguinea*	Con-MB	0.752	0.313	2.397	0.017
*F. sanguinea*	Con-BB	0.834	0.309	2.701	0.007
*F. sanguinea*	MB-BB	0.082	0.271	0.304	0.761
*F. pratensis*	Con-MB	1.946	0.303	6.413	<0.001
*F. pratensis*	Con-BB	2.159	0.301	7.180	<0.001
*F. pratensis*	MB-BB	0.212	0.167	1.267	0.205
*F. truncorum*	Con-MB	0.459	0.179	2.567	0.01
*F. truncorum*	Con-BB	0.751	0.171	4.392	<0.001
*F. truncorum*	MB-BB	0.292	0.155	1.886	0.059
*L. platythorax*	Con-MB	−0.242	0.267	−0.908	0.364
*L. platythorax*	Con-BB	−0.630	0.297	−2.118	0.034
*L. platythorax*	MB-BB	−0.388	0.307	−1.263	0.207
*L. flavus*	Con-MB	−0.974	0.283	−3.441	0.001
*L. flavus*	Con-BB	−0.307	0.228	−1.349	0.177
*L. flavus*	MB-BB	0.667	0.299	2.233	0.026

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
