# Peer review of "Comparison of Twelve Ant Species and Their Susceptibility to Fungal Infection"

_insects, 2019, doi:10.3390/insects10090271_

Round 1
Reviewer 1 Report
The authors have addressed my two main concerns: the fact that their work could not support their conjecture about social immunity and that their videos were too short to capture mutual social interactions known to reduce fungal infection. Hence, their current presentation is more focused and less speculative. Yet, the ms is sprinkled with multiple typos that need fixing before publication can take place. Detailed suggestions are below:
a) Abstract: “compared several facets of the immune defences” ? I disagree with this statement. You basically present one facet: differences in survival. What are the other facets of immune defences? Autogrooming is a behavior not an immune defence.
b) Line 45: typo repeated sentence.. compare with line 42.
c) Throughout the ms, the species names are not consistently in italics.
d) Line 84: add we assumed in “hence, we assume the absence of a queen…”
e) line 138: you could not “ensure”. You could reduce the probability… but really never ensure.
f) Line 212: Should probably read: Autogrooming rates. Behavioral comparisons between control ants and ants exposed to …”
g) Line 219: delete between and instead write “across”. I am surprised that in Fig 4, BB control and BB Blocked by day 10 are not significantly different. They are likely not different from days 0 to 6-7, but by day 10, there is ~a 15% difference in percent survival. Did you use a Breslow or a log rank comparison? Each one of these statistical comparisons gives different weight to the early or later days of your census. I would imagine that by day 10-12, those percent survivals are likely significant and perhaps you chose the statistic that gives higher weight to the earlier days rather than the later? If the results at the end of the census period are significant between these survival curves, then your interpretation of the data would be actually very different.
h) Line 227: Why is this statement noteworthy? Why is it unexpected that different species should exhibit different susceptibilities? What would be surprising is if all these species had similar susceptibilities. How is finding different susceptibilities advancing our understanding of the pathobiology of ants? A deeper assessment of the significance of their results would be welcomes.
Reviewer 2 Report
After carefully reading the revised version and point-by-point responses, I am happy with the manuscript as it is, given that all my comments are properly considered and the text was revised accordingly. Hence, I feel this manuscript meets the standard of the journal Insects and suits well with the special issue, thus recommending Accept as final editorial decision from my side.
Author Response
Please see the attachment

This manuscript is a resubmission of an earlier submission. The following is a list of the peer review reports and author responses from that submission.
Round 1
Reviewer 1 Report
In this manuscript, the authors report on fungal susceptibilities of 12 ant species against 2 types of fungus. Their susceptibility assays, accompanied by behavioral observations and a measure of immune function for each of the ant species, had the potential to make this a nice contribution. Unfortunately, I cannot recommend this work for publication in its present form. Suggestions and what I hope are perceived as constructive criticisms, are given below.
1) The title and the entire presentation of this ms is misleading. I see little evidence relating this work to social immunity. There is really no data here indicating that these different susceptibilities to fungal infection arise from social immunity. Although the authors had the best intention of measuring mutual grooming and trophallaxis as social interactions that may play a role in fungal resistance, the reality is that the filming protocol was inadequate; the duration of the videotaping failed to actually capture the social aspects of disease resistance. The only behavior they could measure was auto-grooming and that is a personal behavior, not a social behavior. Therefore, this research does not fall under the umbrella of social immunity and the work should not be represented as such.
2) Although I applaud the comparative nature of this study, I was disappointed to see the authors completely neglected placing their comparative study within an ecological context. It should not be surprising that all these different species have different outcomes with respect to their susceptibility pathogenic fungus. After all, these different ant species likely evolved under very different selection pressure. While some ants may be arboreal, others may live in soil. Some forage on branches and other on the ground. These are diverse habitats and surely have posed (in evolutionary time) different pathogenic risks. The authors ignored explaining and discussing how differences in the nesting and foraging ecologies correlate with their susceptibilities. A table with this information would have been appropriate. The authors need to really expand on how the observed differences in susceptibility across several ant species make this a novel contribution? Why are these results improving our understanding of the dynamics between social hosts and their pathogens? I don’t think the work, as presented here , makes new contributions to ecological immunology of social insects. Moreover, it really says nothing about social immunity.
Given these two main points, I cannot recommend this work for publication in its present form. The work needs an extensive repackaging. There are additional detailed recommendations/suggestions below:
a) Line 36: what about other behavioral like burial? Cannibalism besides necrophoresis? There is empirical evidence for these behaviors too.
b) Lines 45-47: The authors make a big deal about the fact that most studies only consider a single host species attacked by only one pathogen while measuring only one response variable (survival). Although I applaud their recognition of these significant pitfalls in ecological immunology research, the reality is that the authors fell in one of the same pitfalls they themselves criticized: the measurement of only one immune function (the lysozyme zone of inhibition). I concede that testing 12 different ants species and two fungal pathogens is impressive. However, their behavioral assays provide no real information on how ant social interactions foster social immunity. Although they attempted to measure social behavior, the duration of the videotapes precluded them from actually quantifying ant social interactions. Consequently, they cannot spin their manuscript under the theme of social immunity. There are other incongruences in the ms: the researchers treated ants with fungi yet, they tested for antimicrobial activity in the ants’ hemolymph against a bacterium (Micrococcus)? What is the justification for this? Would it not have been more appropriate to test antifungal rather than antibacterial activity given the ants were treated wiith fungus? Why not measure other aspects of immune function, particularly given the authors' criticisms of other work recording only one response measure? How about measuring antifungal activity? Phenoloxidase activity? Immune-gene expression? Cellular immunity? Rates of phagocytosis? rates of encapsulation? There are ample assays they could have used in addition to the zone of inhibition.
c) What is the justification for attempting to measure throphallaxis? Why is trophallaxis important in “social immunity”. There is actual empirical evidence for this, yet the authors do not even touch on the possible role that trophallaxis has on social immunity. Given they were unable to quantify this behavior, perhaps their attempts (without really getting these data) and my criticisms of the lack of justification for measuring trophallaxis are, mute.
d) More detailed information about temperatures, humidity would be useful. Were the ants fed while on the plaster nests? Are the survival distributions the result of pathogenic+starvation stress? If this was the case, should not the authors at least recognize that the survival in their do not represent susceptibility only to fungal infection but instead represent the combined effects of starvation and disease? Was the age of the workers, or at least their degree of cuticular sclerotization (which has been shown to play an important role in conidia invasion) standardized? I am surprised that they incubated conidia at 37C. Fungus is not bacteria and fungus likes it ~25C, not 37C. Is this temperature a typo? In my experience, fungus does not exhibit the high viability the authors recorded at high temperatures.
e) are all pairwise comparisons followed with a p adjustment? Line 139 does not seem to suggest corrections were made due to multiple pairwise comparisons.
f) why not run survival comparisons using species as a categorical value? In this way, the authors could actually give hazard factors across all species. Their analyses across different treatments seem to be within species, not between species.
g) Line 165: why only the 2 fungi? Why not include controls here too?
h) Graphs could be of higher quality. The vertical line indicating the time at which surviving ants were removed for hemolymph analyses are almost not visible in my copy. Species names in figure legends and references are not in italics.
i) I would have appreciated a photo of the lysozyme experiments to visualize the zones of inhibition
j) The results section would benefit from adding subheadings.
k) Lines 294-298: “Spores” needs to be replaced by “conidia”
l) I worry about the way in which authors confirmed the cause of disease in their exposed ants. Looking at hyphae protruding from the segmental junctions does not in any way assure that the fungus is the same fungus they exposed the ants to. The authors should have allowed the ants to sporulate and then confirm that the conidia are truly exhibiting the tale-tale characteristics of their experimental fungus. In my experience, exposed insects with one pathogen can succumb to other opportunistic pathogens. Their confirmation protocol was cut short too early and therefore, cannot ascertain the ants died of the 2 experimental fungi. I do not remember seeing any data on confirmation rates. Did I missed that? Did control ants ever confirmed positive for fungal infection?
Reviewer 2 Report
The authors have detailed behavioral and physiological responses of a total of 12 ant species to the exposure to 2 fungal pathogens, with a goal to understand species differences in coping with diseases. The role of metapleural gland in defending fungal pathogens is also discussed. This manuscript is well-written and has provided solid, convincing evidence for differential susceptibilities to the fungal pathogens among the tested species and also how the presence of metapleural gland has shaped the evolution of disease defense strategy. I am happy with the manuscript as it is and I do have some comments, mostly in the Introduction section, and feel the study would be improved (thus accepted) if the comments below are properly considered while revising.
In Introduction section, the authors have mentioned several group-level defense strategies, and spent an entire paragraph to explain the metapleural gland and its role in other ant species. I would like to see the authors to explain other strategies and expand more in the Introduction section as an overview, especially for grooming (both auto- and allo-), which happens to be one of the behavioral parameters the authors also have looked into.
I wonder the reason the authors only collected queenless fragments? Would it be different if they carried out the experiments using worker ants from a queenright colony? Please explain the rationale.
Why these 12 species? Any hypothesis-testing framework behind the selection of the 12 species? the presence/absence of the metapleural gland? what are other potential life history traits that may contribute to differences in the susceptibility? What I would like to say is that it would help clarify why these 12 species were sleeted, instead of giving an impression of cherry picking from what was available out there. A great example to refer to is Tranter et al. 2015 Behavioral Ecology in which 4 species with different ecological contexts and life-history traits were compared for the responses to the fugal pathogens.
Line 122-125: the way the authors addressed this part of experiment makes me feel like the authors decided to include the metapleural gland only after they learnt the results of mortality of C. ligniperdus - it would’ve been much better if there is a section that covers hypothesis-testing earlier in the main text.